# Feeding Expressed Breast Milk Alters the Microbial Network of Breast Milk and Increases Breast Milk Microbiome Diversity over Time

**DOI:** 10.3390/microorganisms13010012

**Published:** 2024-12-25

**Authors:** Melissa A. Woortman, Emily S. Barrett, Thomas G. O’Connor, Steven R. Gill, Kristin Scheible, Jessica Brunner, Haipeng Sun, Maria G. Dominguez-Bello

**Affiliations:** 1Department of Nutritional Sciences, Rutgers University, New Brunswick, NJ 08901, USA; melissa.woortman@rutgers.edu; 2Department of Biostatistics and Epidemiology, Environmental and Occupational Health Sciences Institute, Rutgers University, Piscataway, NJ 08854, USA; emily.barrett@eohsi.rutgers.edu; 3Departments of Psychiatry, Neuroscience, and Obstetrics and Gynecology, University of Rochester, Rochester, NY 14642, USA; tom_oconnor@urmc.rochester.edu; 4Department of Microbiology and Immunology, University of Rochester, Rochester, NY 14642, USA; steven_gill@urmc.rochester.edu; 5Department of Pediatrics, University of Rochester, Rochester, NY 14642, USA; kristin_scheible@urmc.rochester.edu; 6Department of Obstetrics and Gynecology, University of Rochester, Rochester, NY 14642, USA; jessica_brunner@urmc.rochester.edu; 7Department of Biochemistry & Microbiology, Rutgers University, New Brunswick, NJ 08901, USA; hs862@sebs.rutgers.edu; 8Department of Anthropology, Rutgers University, New Brunswick, NJ 08901, USA; 9Humans and the Microbiome Program, Canadian Institute for Advanced Research, Toronto, ON M5G 1M1, Canada

**Keywords:** breast milk, breast milk expressing, pumping, microbiota, microbiome

## Abstract

Breastfeeding supplies nutrition, immunity, and hormonal cues to infants. Feeding expressed breast milk may result in de-phased milk production and feeding times, which distort the real-time circadian cues carried by breast milk. We hypothesized that providing expressed breast milk alters the microbiotas of both breast milk and the infant’s gut. To test this hypothesis, we analyzed the microbiota of serial breast milk and infant fecal samples obtained from 14 mother–infant dyads who were lactating, half of which were providing expressed breast milk. Infant fecal microbiota showed lower α-diversity than breast milk microbiota. Bacterial amplicon sequence variant sharing occurred between breast milk and infant feces with no feeding group differences. However, the age-dependent gain in breast milk α-diversity was only significant in the expressed breast milk group and not in the direct breastfeeding group, suggesting that decreased contact with the infant’s mouth influences the milk microbiota. Trending lower connectivity was also noted with breast milk microbes in the direct breastfeeding group, consistent with regular perturbations of the developing baby’s oral microbiota by latching on the breast. The results of this preliminary study urge further research to independently confirm the effects of providing expressed breast milk and their health significance.

## 1. Introduction

The gut microbiome—a symbiotic community of microbes that exists in the gastrointestinal tract of mammals—starts assembling during birth, when the first colonizers from the birth canal and perineum arrive in the neonate gut [1]. There has been a debate over the sterility of the healthy fetus and placenta, but recent work that was carefully performed with the needed controls has challenged the notion of the placental microbiome, showing a contamination source of the observed bacterial DNA, which in the opinion of many in the microbiome field (including us), has resolved the controversy [2,3,4,5,6]. After this initial inoculation, the first year of life is when the gut microbiome is most pliable, and perturbations during this critical developmental window can predispose infants to inflammatory conditions, such as obesity, later in life [7,8,9]. Certain early life practices, such as antibiotics, cesarean section birth, and lack of breastfeeding, have been found to increase weight gain and fat deposition [7,10]. Furthermore, formula feeding accelerates the maturation of the gut microbiome [11,12] and correlates with accelerated growth, increasing obesity risk later in life [13,14].

In addition to feeding the infant, breast milk promotes the growth of certain microbes in the intestine, such as beneficial species of *Bifidobacterium*. Particularly, *Bifidobacterium bifidum*, *Bifidobacterium breve*, and *Bifidobacterium longum* subspecies *infantis* can preferentially utilize human milk oligosaccharides (HMOs), prebiotic short-chain carbohydrates found in human breast milk, as a carbon source to increase growth. A higher abundance of these bacteria species during infancy is associated with decreased risk of developing autoimmune diseases due to their many beneficial metabolites, such as the production of beneficial short-chain fatty acids [15]. Breast milk also provides growth-modulating hormones such as cortisol, leptin, and melatonin [16,17,18], some of which fluctuate in breast milk throughout the day, mimicking their circadian concentrations in maternal serum [19,20]. Furthermore, lactoferrin and IgA are two compounds in breast milk that provide beneficial effects for infant immunity, including protecting the infant from infection by potential pathogens [21,22]. Breast milk lactoferrin and IgA have been found to change in concentration in response to illness in the infant [23].

In one U.S. study, up to 85% of mothers in urban areas reported having expressed breast milk, while about two-thirds of lactating mothers with infants aged 4.5–9.5 months reported regularly feeding their babies expressed breast milk [24]. Bottle-feeding expressed breast milk, an evolutionary novelty, may present a physiologic stressor, exposing the infant to “mis-timed” hormones and other bioactive factors (i.e., due to differences between the time of milk production and consumption). The practice of expressing breast milk can subject the milk to fluctuating temperatures during storage and rewarming, which can alter its composition [25,26,27]. This process may affect the concentration and functionality of various nutrients and bioactive components in breast milk, with the extent depending on storage duration, temperatures reached, and time held at high or low temperatures [28,29]. Moreover, a common practice with expressed breast milk is combining milk from several pumping sessions, or pooling, which can negate circadian hormonal cues. Providing expressed breast milk may also lead to early cessation of breastfeeding [30,31]. We hypothesized that bottle-feeding expressed breast milk alters the breast milk and infant gut microbiotas compared to direct breastfeeding. Evaluating these potential gut microbiome stressors may help to explain how the gut microbiome modulates growth and a healthy BMI, and how to mitigate an increased risk of obesity later in life for infants.

## 2. Materials and Methods

### 2.1. Study Design

Subject data and samples used for this study were from the Understanding Pregnancy Signals In Development (UPSIDE) cohort [32]. Pregnant women meeting the UPSIDE study criteria (Appendix A) were recruited for an observational, longitudinal study that followed participants until their child reached 4 years of age. For this secondary analysis, mother–infant dyads were selected from the larger UPSIDE cohort based on the following criteria: sufficient samples available, breastfeeding status, and absence of known microbiome disruptions (e.g., cesarean section birth, formula feeding, or antibiotic use) based on medical records and maternal reports. The selected mother–infant dyads were classified into two groups according to the frequency of consistently feeding expressed breast milk. Infants in the expressed breast milk group had received two or more bottles of breast milk during maternal workdays after one month postpartum (according to maternal report), whereas the direct breastfeeding group had provided fewer than two bottles of expressed breast milk to their infants per maternal workday. As part of the UPSIDE study protocol, mothers reported the number of breast milk bottles given to their infant over a typical 24 h period on a workday. The mothers in this analysis were still providing breast milk to their infants at the six-month time point, so cessation of breastfeeding had not occurred, and the infants may have started on solid foods prior to reaching six months of age. This study was approved by the Rutgers University New Brunswick Health Sciences Institutional Review Board (Protocol # Pro20160001514, approval date 27 January 2017) and the University of Rochester Research Subjects Review Board (Protocol RSRB:00058456, approval date: 27 August 2015).

### 2.2. Sample Collection

Breast milk samples from the mothers and fecal samples from the infants were collected at 1 month and 6 months postpartum. Infant feces were collected by the mothers from infant diapers by using provided tubes and supplies prior to freezing. Collection of breast milk occurred either at home or during the study visit using either the mother’s own pump supplies or supplies provided by the study team. Timing of collection varied based on whether the milk was collected at home or at a study visit, which was usually before noon. Mothers provided approximately 30 mL of breast milk from one or both breasts, which was then aliquoted into cryotubes by the study team prior to storage. Biological samples were transported to Rutgers University on dry ice and stored at −80 °C until analysis.

### 2.3. DNA Extraction and Sequencing

Breast milk samples were thawed and then vortexed briefly, and 500 μL of breast milk from each sample was centrifuged at 15,000× *g* at 4 °C using the Eppendorf Centrifuge 5810 R (Eppendorf AG, Hamburg, Germany). The supernatant and fat layers from the samples were then removed and the DNA was extracted from the pellet. Infant fecal samples were thawed, and 50 mg was used for DNA extraction. Breast milk DNA was extracted using the Qiagen DNeasy Powersoil Pro Kit (Qiagen, Hilden, Germany), and fecal DNA was extracted with the Qiagen DNeasy Powersoil HTP 96 Kit (Qiagen, Hilden, Germany). Quantification of the extracted DNA concentration was performed via the Quant-iT PicoGreen dsDNA assay kit (Thermo Fisher Scientific, Eugene, OR, USA) for the fecal samples and via the Qubit dsDNA HS assay kit (Thermo Fisher Scientific, Eugene, OR, USA) for the breast milk samples.

Polymerase chain reaction (PCR) was carried out for both sample types using the Invitrogen Platinum Hot Start Master Mix (Thermo Fischer Scientific, Vilnius, Lithuania) to amplify the V4 variable region of 16S ribosomal ribonucleic acid (rRNA) from the bacterial genes according to the Earth Microbiome Project [33]. The primers for amplification are FWD:GTGYCAGCMGCCGCGGTAA; REV:GGACTACNVGGGTWTCTAAT. Negative controls for DNA extraction and PCR were run along with the breast milk and fecal samples. The DNA concentration of the PCR products was quantified via the Quant-iT PicoGreen dsDNA assay kit (Thermo Fisher Scientific, Eugene, OR, USA). Equimolar amounts of the DNA from each sample were pooled and purified with the Qiagen QIAquick PCR purification kit (Qiagen, Hilden, Germany). The samples were sequenced using Illumina MiSeq at GENEWIZ, LLC (South Plainfield, NJ, USA).

### 2.4. Microbiome Analysis

Sequence reads were demultiplexed and denoised using the QIIME2 pipeline (v2023.5) [34], reads were clustered into amplicon sequence variants (ASVs) with DADA2 [35], and taxonomic assignment of the ASVs was aligned with the SILVA database v132 [36]. The outputs were imported into RStudio (v4.3.1) [37] to remove possible contaminants and non-microbial sequences using the decontam package [38]. Microbiome data were rarified to a sampling depth of 871 prior to generating diversity metrics (Appendix A) to ensure that no samples were filtered out. Alpha diversity metrics (Faith’s phylogenetic diversity, Shannon Index, Pielou’s Evenness, and observed features) were calculated to assess microbial diversity within samples. Beta diversity, which compares microbial diversity between samples, was also estimated using Jaccard, Bray–Curtis, and weighted/unweighted UniFrac distance matrices. Microbiomes were also compared using Analysis of Compositions of Microbiomes with Bias Correction (ANCOM-BC) [39] to determine the differential abundances between groups. To look at microbe sharing between breast milk and infant feces, unique ASVs were considered to be shared if they were present in both samples from the mother–infant dyad. Finally, a co-abundance network analysis was conducted to examine bacterial abundance changes by identifying increases and decreases in ASV abundance and exploring their interrelationships. Bacteria that increase or decrease in abundance together suggest beneficial symbiosis, whereas an inverse relationship indicates potential competition or antagonism [40]. This analysis was completed by using the SpiecEasi package in RStudio [41], with correlation cutoff = 0.3 and p value cutoff = 0.05.

### 2.5. Statistical Analyses

Normality was determined by using the Shapiro–Wilk test. For normally distributed data, we used the student *t*-test, and otherwise used the non-parametric Wilcoxon Rank Sum Test. Subject characteristics were examined either using the student *t*-test or Wilcoxon Rank Sum Test to compare numerical value medians, and either Fischer’s exact test or chi-square test to compare categorical values that had two comparisons or three or more comparisons, respectively. Wilcoxon Rank Sum Test was used to compare α-diversity, and significance for β-diversity was calculated using permutational multivariate analysis of variance (PERMANOVA). For ASV sharing between infant feces and maternal breast milk, chi-square test was used to compare the number of shared ASVs, as well as the number of mother–infant dyads that shared ASVs between the direct breastfeeding and expressed breast milk groups. Wilcoxon Rank Sum Test was used to compare the average degree of connectivity in co-abundance networks with the two breastfeeding groups. Statistical analyses were carried out using the base package of RStudio, except for PERMANOVA, which utilized the vegan package in R [42].

## 3. Results

Out of the 327 mother–infant dyads in the UPSIDE cohort that were enrolled at the time of this study, 313 were excluded based on the criteria for this analysis. The remaining 14 mother–infant dyads were included in this study—six mother–infant dyads in the expressed breast milk arm and eight in the direct breastfeeding arm (Figure 1). The two groups were similar in terms of most baseline characteristics (Table 1 and Table 2). Employment was significantly lower among the direct breastfeeding group mothers when compared to the expressed breast milk group before becoming pregnant, as well as at six months postpartum, which was expected as mothers often express breast milk to be fed via bottle due to their work schedule. All six mothers in the expressed breast milk group reported working 40 h/week. The infants did not vary significantly in terms of baseline characteristics. The expressed breast milk group all used childcare at 6 months of age, either at home with someone other than a parent watching the infant, or at a daycare. The age for starting solid foods was significantly higher in the direct breastfeeding group compared to the expressed breast milk group.

### 3.1. Microbiome Differences Between Breast Milk and Infant Feces

Prior to separating by breastfeeding mode, aggregate breast milk and infant fecal microbiota were compared. Bacterial alpha diversity was higher in maternal breast milk than in the infant fecal microbiome (Figure 2A and Appendix A), with beta diversity also differing between breast milk and infant fecal microbiomes (Appendix A). In terms of differential ASV abundance, breast milk had a higher proportion of typical skin bacteria such as *Streptococcus* sp., and infant feces had typical fecal bacteria such as *Bacteroides*, *Enterococcus*, *Veilonella*, *Intestinibacter*, and *Clostridiodes difficile* (Figure 2B).

### 3.2. Age-Related Differences of Infant Fecal and Breast Milk Microbiomes

Breast milk microbiota changed with the age of the baby, with significant differences in beta diversity (Appendix A), although not in alpha diversity (Appendix A). From 1 to 6 months post-partum, there was a significant enrichment of *Streptococcus* sp. in maternal milk noted using ANCOM (Appendix A).

Infant fecal microbiota’s alpha diversity increased with age from 1 to 6 months for Faith’s Phylogenetic Diversity and Observed Features (Appendix A), with age-dependent increases in the relative abundance of *Veilonella* sp., *Clostridiodes difficile*, and *Intestinibacter* sp. (Appendix A). Beta diversity was not significantly different between ages for infant fecal microbiota, but a trend towards significance (*p* = 0.053) was noted with unweighted Unifrac (Appendix A).

### 3.3. Breastfeeding Mode and the Microbiomes of Breast Milk and Infant Feces

When comparing microbiomes based on feeding mode, there were no differences in alpha or beta diversity between breastfeeding modes in the microbiome of maternal breast milk (Appendix A) or infant feces (Appendix A). No significantly discordant taxa were found at six months between the feeding groups when using ANCOM to detect discordant ASVs between breastfeeding modes.

When comparing sample types while separating by feeding mode, the expressed breast milk group had more significant differences in alpha diversity between the microbiomes of infant feces and maternal breast milk than the direct breastfeeding group (Appendix A). Compared to directly breastfeeding mothers, mothers who provided their infants with expressed breast milk showed an accelerated gain in breast milk alpha diversity with time post-partum (Figure 3 and Appendix A), while a non-significant trend was observed in the infant fecal microbiome (Appendix A).

There was substantial bacterial sharing between breast milk and infant feces (Figure 4), including taxa in four phyla, namely Firmicutes (*Veillonella* sp., *Gemella* sp., *Streptococcus* sp., *Schaalia odontolytica*), Actinobacteria (*Bifidobacterium* sp., *Rothia mucilaginosa*), Proteobacteria (*Enterobacteriaceae* sp., *Escherichia-Shigella* sp., *Acinetobacter* sp., *Haemophilus* sp.), and Bacteroidetes (*Bacteroides fragilis*). There were no significant differences in the number or prevalence of shared ASVs in maternal breast milk and infant feces when comparing feeding groups.

Co-abundance network analysis of the breast milk microbiota revealed a large central cluster in the expressed breast milk group, which included a few *Streptococcus* species, *Bacillus*, *Granulicatella elegans*, *Veillonella*, and *Haemophilus*. This indicates that these taxa play a more influential role in driving changes in bacteria abundance within the cluster (Figure 5A). Connectivity differences between the expressed breast milk and direct breastfeeding groups showed a trend towards significance (*p* = 0.082), suggesting that the microbial communities in the expressed breast milk group may be more interconnected in terms of bacterial abundance changes. In the direct breastfeeding group, the breast milk co-abundance network analysis displayed more dispersed clusters centered around *Streptococcus*, *Acinetobacter*, and *Neisseria*. The infant fecal co-abundance network showed fewer, more closely connected clusters in the direct breastfeeding than in the expressed breast milk group. However, no significant differences in connectivity were observed (Figure 5B). The mostly negative correlations in the networks indicate that as the abundance of one taxon increases, the abundance of another decreases, indicating competition, niche exclusion, or the production of metabolites that inhibit the growth of other taxa.

## 4. Discussion

This study compared bacteria from maternal milk and infant feces at one and six months after birth to investigate differences between feeding expressed breast milk and directly breastfeeding only. We found that infant feces had lower alpha diversity than breast milk prior to separating by breastfeeding mode. These results are consistent with other studies that have compared infant fecal and breast milk microbiomes [43,44]. Low gut microbiome diversity is seen as beneficial for infants prior to weaning off of breast milk due to the influence of the gut microbiome on maturing biological systems, such as the immune system, during this time [45]. Interestingly, at six months, the differences in observed features, Shannon Entropy, and Faith’s Phylogenetic Diversity were only significant for the expressed breast milk group when separated by feeding mode. When feeding expressed breast milk, the infant is often fed by someone who is not the mother. In this study, all of the infants in the expressed group were either cared for by a family member at home or in a daycare setting. The lack of direct breastfeeding and the presence of other caretakers could have an effect on the infant microbiome [46] and, therefore, decrease the similarity between it and the maternal breast milk microbiome. Additionally, breast milk that is expressed often undergoes storage and warming, which could alter the components in the milk that is fed to infants [26,47]. The effect of subjecting breast milk to temperature extremes, such as heating over 62 °C [26], may also contribute to differences in the fecal microbiomes of those infants being fed expressed milk.

Previous studies have shown no change in alpha diversity in breast milk after one month postpartum [43,48], which is consistent with our aggregate results. However, when separated by feeding mode, only the expressed breast milk group had significant differences between one and six months postpartum for alpha diversity. These results suggest that the maternal milk microbiome may also be affected by decreased direct contact of the maternal breast with the infant’s mouth.

Other studies, such as those from the CHILD Cohort project in Canada, have reported the sharing of bacteria between infant feces and maternal breast milk varying by breastfeeding mode, with a higher proportion of >5 or more ASVs shared with mother–infant dyads directly breastfeeding versus those who provided some expressed milk [49]. The CHILD Cohort researchers also observed that breast milk feeding practices were associated with some alpha diversity differences, beta diversity differences, and bacterial sharing in the mother’s milk and infant feces at three months after birth [50]. However, these analyses included mother–infant dyads subjected to previously established microbiome stressors, such as cesarean section birth [7,8], antibiotic usage [9,51], and formula feeding [52,53], which could have a synergistic effect on the infant’s gut when expressed breast milk is provided via a bottle.

Co-abundance networks determine connectivity and direction of correlations in the relative abundance of different bacterial taxa, and a high connectivity node typically reflects a more stable consortium that is less susceptible to change [54]. Bacterial taxa at the center of the connectivity nodes will have more influence on microbiome composition, as changes in abundance of highly connected taxa will likely result in abundance changes of the other taxa in that node. The results of this study show that there was a trend in significance towards higher connectivity in the breast milk of the expressed breast milk group. Higher connectivity in the breast milk microbiota might be related to less frequent exposure of the maternal breast and infant oral cavity in the process of providing expressed breast milk. The practice of direct breastfeeding is an opportunity for the oral-to-breast transmission of microbes [55,56]; therefore, exposing the breast to the microbes in the infant’s oral cavity less frequently could decrease the instances of bi-directional transmission [57]. The study showed that for the infant feces, it was the direct breastfeeding group that had potentially higher connectivity, as demonstrated by two clusters versus three in the expressed group. The fecal co-abundance might also reflect the lower perturbations of the infant gut after feeding naturally at the same time when milk is produced, and consuming breast milk directly from the breast is conducive to better synchrony of hormones and activity–sleep patterns with the mother [58].

A limitation of this study is the small number of mother–infant dyads utilized for the analysis. Another possible limitation is maternal employment as a confounder since expressing breast milk often occurs as a result of the mother returning to work. Along with differences in maternal work and infant care environments, other environmental exposures and potential contamination of breast milk pumping and storage supplies could have contributed to the microbiota differences noted in this study. Future studies should consider microbial sources, such as the mother’s breast skin, bottles, breast milk pumping supplies, and the environmental surfaces the milk is exposed to, in order to control for potential contamination. The initiation of solid food also influences the infant gut microbiome [46,59], and significant differences were observed between the two groups regarding the average age at which solid foods were introduced. This finding aligns with previous research linking feeding expressed breast milk and maternal employment to a shorter duration of breastfeeding [60,61,62]. Introducing solid foods earlier may reduce the reliance on breast milk as a primary nutrient source and contribute to earlier breastfeeding cessation.

While breast milk remains the gold standard for infant feeding, the results of this study show that there may be differences in the breast milk and infant fecal microbiomes based on breast milk feeding mode. These differences in the microbiomes of breast milk and infant feces deserve further confirmation and research to establish the mechanisms and health significance of such differences, particularly given the high prevalence of expressing breast milk in some countries.

## Figures and Tables

**Figure 1 microorganisms-13-00012-f001:**
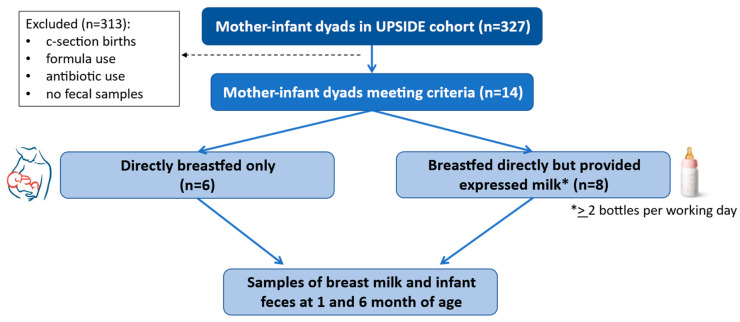
Study design. Mother–infant dyads from the UPSIDE cohort were selected for this analysis if they met the described criteria. Once selected, the dyads were placed in either the directly breastfed only group or the group that breastfed directly but provided some expressed milk, with the latter being defined as feeding at least two bottles of expressed breast milk per working day.

**Figure 2 microorganisms-13-00012-f002:**
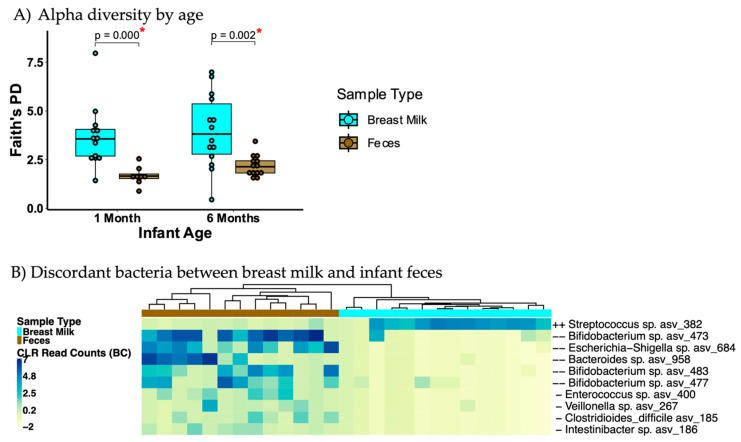
Microbiota differences between breast milk and infant feces. (**A**) Alpha diversity at one month and six months of age with each sample represented as a point in the boxplot. (**B**) Unsupervised clustering of read counts of significantly discordant amplicon sequence variants (ASVs) between breast milk and feces of 6-month-old infants. When comparing alpha diversity medians, breast milk showed higher alpha diversity than infant feces at 1 and 6 months of age; breast milk has higher *Streptococcus* and lower *Bifidobacterium*, *Escherichia*, and *Bacteroides*. Each sample is represented as a point in the boxplot for alpha diversity. Alpha diversity differences were tested using Wilcoxon Rank Sum Test. Discordant bacteria determined using Analysis of Compositions of Microbiomes with Bias Correction (ANCOM-BC); the figure shows log fold change: ++: ≥2 Log fold change higher in breast milk; -: 1–2 Log fold change lower in breast milk; --: ≥2 Log fold change lower in breast milk. CLR = Central Log transformation Ratio; BC = Bias Corrected. * indicates significant *p* value of <0.05.

**Figure 3 microorganisms-13-00012-f003:**
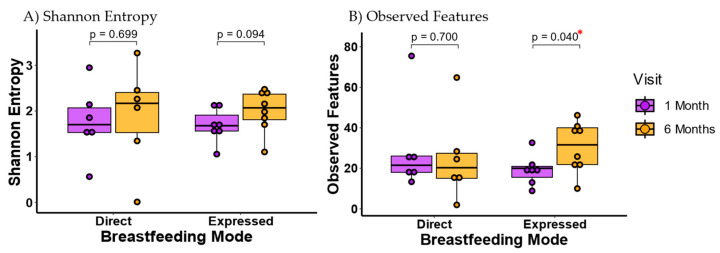
Bacterial alpha diversity in breast milk between age groups by breastfeeding mode. (**A**) Shannon Entropy. (**B**) Observed features. All infants at 1 month were directly breastfed, and providing expressed breast milk occurred between 1 and 6 months only in the expressed breast milk group. Each sample is represented as a point in the boxplot. Alpha diversity differences were tested using Wilcoxon Rank Sum Test. Breast milk microbiota alpha diversity showed a trend to increase with age, which was only significant in the expressed breast milk group. * indicates significant *p* value of <0.05.

**Figure 4 microorganisms-13-00012-f004:**
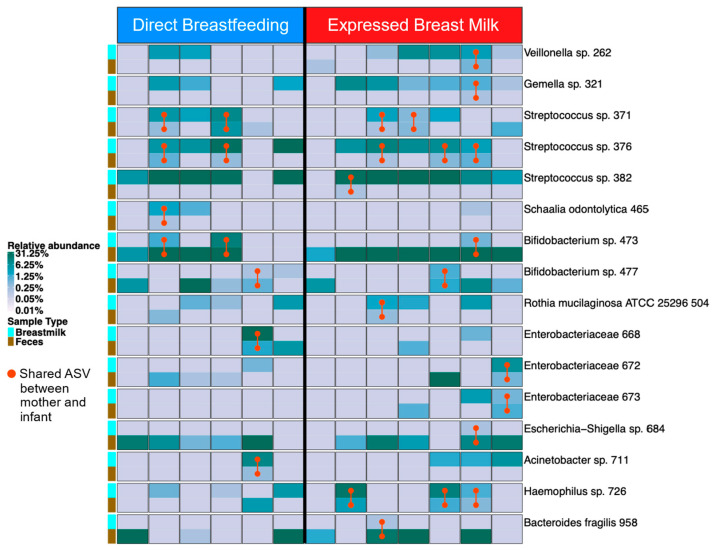
Relative abundance in breast milk bacterial taxa and in infant feces in the two feeding groups at 6 months postpartum. Red vertical lines indicate shared ASV between breast milk and infant feces. No significant group differences were found in the number or prevalence of shared ASVs. Comparisons made using chi-square test. ASV = amplicon sequence variant.

**Figure 5 microorganisms-13-00012-f005:**
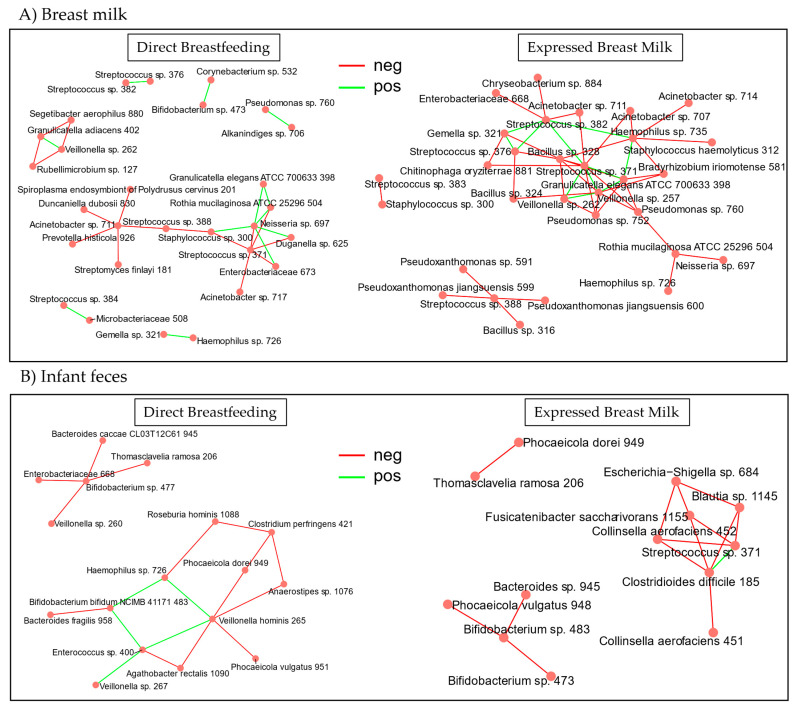
Microbiota co-abundance networks in breast milk feeding mode at 6 months after birth. (**A**) Co-abundance network of breast milk microbiota in each feeding mode. (**B**) Co-abundance network of infant fecal microbiota in each feeding mode. Connectivity differences between breast milk bacteria of the expressed breast milk group trended towards significance (Wilcoxon Rank Sum test, *p* = 0.082) based on the average connections of the amplicon sequence variants (ASVs), and for the fecal bacteria, there were no significant differences in connectivity (Wilcoxon Rank Sum test, *p* = 0.877). neg = negative correlation, pos = positive correlation.

**Table 1 microorganisms-13-00012-t001:** Characteristics of the maternal study subjects. Wilcoxon Rank Sum Test was used to compare characteristics shown as mean values; chi-square test was used to compare characteristics shown with n values. DBF = Direct breastfeeding; EBM = Expressed breast milk. * indicates significant *p* value of <0.05.

Maternal Subject Characteristics	DBF (n = 6)	EBM (n = 8)	*p* Value
Age, years (mean ± sd)	31.5 ± 3.62	30.5 ± 1.41	0.948
Pre-Pregnancy BMI, kg/m^2^ (mean ± sd)	21.2 ± 3.84	23.4 ± 4.33	0.573
Gestational Weight Gain, kg (mean ± sd)	11.0 ± 7.87	13.1 ± 4.38	0.175
Annual Household Income (mean ± sd)	$72,367 ± 42,107	$122,125 ± 49,875	0.106
Race (n, %)			0.239
White	6, 100%	5, 62.5%	
Asian/Pacific Islander	0, 0%	2, 25%	
Black/African American	0, 0%	1, 12.5%	
Ethnicity (n, %)			0.429
Hispanic	1, 16.7%	0, 0%	
Non-Hispanic	5, 83.3%	8, 100%	
Marital Status (n, %)			0.429
Married	5, 83.3%	8, 100%	
Single	1, 16.7%	0, 0%	
Education Level (n, %)			0.626
High School Diploma	1, 16.7%	0, 0%	
Some College	1, 16.7%	1, 12.5%	
Bachelor’s Degree	1, 16.7%	1, 12.5%	
Post-Graduate Degree	3, 50%	6, 75%	
Employed Pre-Pregnancy (n, %)			0.015 *
Yes	2, 33.3%	8, 100%	
No	4, 66.7%	0, 0%	
Employed 6 Months Postpartum (n, %)			0.003 *
Yes	1, 16.7%	8, 100%	
No	5, 83.3%	0, 0%	
Alcohol Use Pre-Pregnancy (n, %)			1.00
Yes	4, 66.7%	6, 75%	
No	2, 33.3%	2, 25%	
Smoking Pre-Pregnancy (n, %)			0.429
Yes	1, 16.7%	0, 0%	
No	5, 83.3%	8, 100%	
Drug Use Pre-Pregnancy (n, %)			0.429
Yes	1, 16.7%	0, 0%	
No	5, 83.3%	8, 100%	
Breast Milk Bottles/Workday 6 Months Postpartum (n, %)			0.007 *
0	5, 83.3%	0, 0%	
1	1, 16.7%	0, 0%	
2	0, 0%	3, 37.5%	
3	0, 0%	4, 50%	
≥4	0, 0%	1, 12.5%	

**Table 2 microorganisms-13-00012-t002:** Characteristics of infant subjects in the study. Wilcoxon Rank Sum Test was used to compare characteristics shown as mean values; chi-square test was used to compare characteristics shown with n values. DBF = Direct breastfeeding; EBM = Expressed breast milk. * indicates significant *p* value of <0.05.

Infant Subject Characteristics	DBF (n = 6)	EBM (n = 8)	*p* Value
Gestational Age, weeks (mean ± sd)	39.7 ± 1.06	40.1 ± 1.02	0.364
Birth Weight, kg (mean ± sd)	3.46 ± 0.47	3.56 ± 0.55	0.755
Birth Length, cm (mean ± sd)	53.45 ± 2.30	50.86 ± 1.78	0.741
Birth Weight-for-Age, z-score (mean ± sd)	0.36 ± 0.95	0.42 ± 1.05	0.852
Birth BMI, z-score (mean ± sd)	−1.13 ± 0.76	0.18 ± 1.44	0.093
Sex (n, %)			0.091
Female	4, 66.7%	1, 12.5%	
Male	2, 33.3%	7, 87.5%	
Number of Siblings (n, %)			0.196
0	2, 33.3%	3, 37.5%	
1	1, 16.7%	3, 37.5%	
2	3, 50%	0, 0%	
≥3	0, 0%	1, 12.5%	
Unknown	0, 0%	1, 12.5%	
Childcare Type at 6 Months (n,%)			0.001 *
At home only	0, 0%	4, 50%	
Daycare	0, 0%	4, 50%	
N/A	6, 100%	0, 0%	
Feeding Solid Food at 6 Months (n,%)			0.429
Yes	5, 83.3%	6, 100%	
No	1, 16.7%	0, 0%	
Age Started Solid Foods, months (mean ± sd)	5.4 ± 0.55	4.375 ± 0.74	0.030 *

## Data Availability

The Illumina 16S rRNA V4 amplicons generated in this study were deposited in the Sequence Read Archive under BioProject PRJNA1186977.

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
