# Peer review of "Feeding Expressed Breast Milk Alters the Microbial Network of Breast Milk and Increases Breast Milk Microbiome Diversity over Time"

_microorganisms, 2024, doi:10.3390/microorganisms13010012_

Round 1
Reviewer 1 Report
Comments and Suggestions for Authors
This is an interesting attempt to address the questions raised on the importance of infants feeding with breast milk. While this, particularly with respect to the circadian rhythms of breast milk composition is an important issue, there remain important questions:
Major:
General: Define the terms alpha- and beta-diversity in the introduction.
L. 46: Don't agree! The fetus and swallowed amniotic fluid are not sterile, and the first microbiota develops in utero.
L.87ff: Less than 2/more than 2 isn't absence or presence.
L.96ff: State that the infants were identically fed solely with breast milk for 6 months here, or specify conditions.
L.137: Specify/declare that data are normally distributed or use non-parametric tests.
L152ff: This is Methods! Provide a flow diagram, in the main body of the text, defining the reasons for and numbers of excluded dyads.
Table 1: not sharp; provide as a text rather than foto. Legend to Table is missing.
Figure 1: Fig. 1A: Use black symbols to be discernable from the boxisFig. Provide information that these are medians, IQRs and individual data points. Data clearly show that they are not normally distributed. Use adequate test (see above, Statistics). 1B isn't readble, as symbols/letters are too small. The lines on top are much too small as well. Re-design!
Fig 2: Low quality figure. Letters partially much too small, unsharp. It isn't clear from the figuer, how many data points are included. Specify in the legend. Use black dots. What's 0.xxx? Define them as - I suppose - p-values in the legend.
Fig. 3: Poor figure design. Fonts too small, unsharp, style looks like a screen shot. Colorful design isn't necessarily a good design.
Fig. 4: Irrespective of generating a sharp figure, the fonts are too small and require substantial re-editing!
L189ff: Figs S5-7: Why are these most important data in the supplement?
L.198f: This higher diversity may simply be due to the contamination by bottles etc. While this is, together with different living conditions (work vs. home; differences in starting solid foods) addressed under limitations, it is a major confounder of the whole study. Excluding the impact of mother's beast skin, the microbes of milk bottels, suckling equipment would have been an essential part of study concept. In fact, the whole study just contributes to the effects of direct vs. bottle feeding of mother's milk.
Minor:
L35: >suggesting accelerated infant gut microbiota maturation.<. Of which group?
L.50: What's meant by growth here?
L. 53: Define these 'certain microbes', like you did for the hormones.
L. 57: Define 'beneficial effects'.
L.103: Provide centrifugal force (*g), time, temperature and model of centrifuge
L185ff: Is this true in mean or does it apply to all individual dyds as well, showing the same tendency?
L.254f: Is it the presence of persons or their microbiota's impact? Specify.
L.257: Define the temperature extremes in breast milk routine handling.
Reviewer 2 Report
Comments and Suggestions for Authors
Interesting study that shows the relationship between bacterial diversity in breast milk (BM) and neonatal feces at 2 ages and related to 2 nutritional practices, direct breastfeeding (DBF) and bottle feeding. The authors demonstrate the relationship between bacterial taxa between these practices, as well as the decrease in diversity in neonatal feces. These are interesting data, however, some contributions would improve the study and its understanding.
- Introduction. Concise and well focused, the objective of the work is correctly understood and the concepts that are going to be explored are explained.
- Material and methods. Are the inclusion and exclusion criteria from the UPSIDE study? Lines 83-85 appear to be the inclusion criteria for this study. Please clarify. The study flow chart should be in the main text, not in the supplementary material.
- What does "working day" mean or imply in this study?
- How was the BM extraction? time of day, both breasts, what volume was collected? On the other hand, when were neonates on DBF at 6 months of age?
- Results. The tables could have a larger font size to make them more readable. On the other hand, table S2 has important characteristics to be incorporated in table 1.
- On the other hand, it is interesting that solids began to be introduced at an older age in the DBF group, which could be related to work-family balance.
- Figure 4. It is difficult to interpret.
- Discussion. I very much agree with the limitations of the study, both the sample size and the effect of the introduction of solids can affect the diversity between the groups analyzed.
- What differences does this study have with references 33 and 34? This study suggests that maintaining bacterial diversity in the newborn at low levels can contribute to better infant health outcomes. Is this due to moderate weight gain and fat accumulation? why?
- The conclusions of the study are not clear.
Round 2
Reviewer 2 Report
Comments and Suggestions for Authors
Thank you forma this revised version.